# AnomalyTCN: Dual-branch Convolution with Contrastive Representation for Efficient Time Series Anomaly Detection

## Abstract

This paper focuses on the rising contrastive-based method for time series anomaly detection, which works on the idea of contrastive discrepancy learning and breaks through the performance bottleneck of previous reconstruction-based methods. But we also find that, existing contrastive-based methods only work with the complicated attention mechanisms, which brings heavier computational costs. To address this efficiency issue, we propose **AnomalyTCN** as a more efficient and effective contrastive-based solution. In detail, we design a dual-branch convolution structure to produce different representations of the same input under two different views for contrastive learning. Then we adopt the representation discrepancy between these two branches as a more distinguishable criterion to detect the anomalies, leading to better detection performance. Meanwhile, since we adopt a simple and light-weight pure convolution structure to avoid the complicated attention computation, our method can enjoy much more advantages in efficiency. Experimentally, our AnomalyTCN achieves the consistent state-of-the-art performance on various time series anomaly detection tasks while saving 83.6% running time and 20.1% memory usage. These results validate that our AnomalyTCN is a novel solution for time series anomaly detection with a better balance of performance and efficiency.

## 1 Introduction

Time series anomaly detection is widely used in extensive real-world applications, including industrial monitoring, financial fraud detection and system maintenance (Anandakrishnan et al., 2017; Cook et al., 2020; Golmohammadi & Zaiane, 2015; Zhong et al., 2023; Ren et al., 2019). In real-world applications, various large-scale time series data are collected during the running time. And time series anomaly detection aims to detect the abnormal time points from them, therefore having great significance for ensuring security and avoiding financial loss. And in practice, the anomalies are rare and hidden by vast normal points, making it hard and expensive to label the data (Xu et al., 2021). Thus, this paper is dedicated to proposing an unsupervised method for time series anomaly detection.

Unsupervised time series anomaly detection is still an open question and followings are some key challenges that need to be resolved. First, a powerful backbone and a non-trivial learning task should be designed to learn informative representation from diverse input series in an unsupervised manner. Second, a distinguishable criterion should be derived to detect the rare anomalies from plenty of normal time points. Recently, various time series anomaly detection methods are developed to handle these challenges. Among them, the classic reconstruction-based methods (Wang et al., 2023; Li et al., 2021; Su et al., 2019) are most popular. They adopt the reconstruction task during the training phase, where ideally a model is trained to only reconstruct the normal samples. Then in testing phase, this model will fail to reconstruct the abnormal samples and thus can detect the anomalies by larger reconstruction errors. **However, such reconstruction-based methods are less-robust for they are easily interfered by the uncleanliness of training data**. In detail, the effectiveness of these methods is heavily related to the assumption that the model is not trained on any abnormal samples during the training process. And the key to this assumption is that we have to ensure the training data is clean and only contains normal points. Otherwise, the model will also learn to reconstruct the abnormal samples and thus fail to produce larger reconstruction errors for anomalies, making them hard to be distinguished. But in time series anomaly detection where the normal and abnormal points may

appear in one instance, the training data is unavoidable to contain some unexpected anomalies, hence disrupting the training process in reconstruction-based methods and limiting their performance.

By contrast, the rising contrastive-based method is more robust to training data. This is because it takes the idea of contrastive discrepancy learning (Chen et al., 2020a; He et al., 2019) and works based on a finding about the representation discrepancy, which is less related to the training process. And this better robustness contributes to the performance superiority of contrastive-based methods. Taking the previous state-of-the-art method DCdetector (Yang et al., 2023) as an example, it proposes a specialized dual-attention structure for contrastive discrepancy learning. When given an input sample, it calculates two attention maps from two different views and compares them. With its finding that the representation discrepancy of two attentions will be larger when faced with anomalies instance, DCdetector adopts this discrepancy as a more distinguishable criterion and achieves better anomaly detection performance. Since this finding is not related to the training process, DCdetector is more robust to training data and surpasses previous reconstruction-based methods by a large margin.

Although achieving ideal performance, the dual-attention in DCdetector brings severe computational costs due to its quadratic complexity. **To improve efficiency, we need to design an attention-free contrastive-based method**. And designing a pure convolution structure for it is a good idea. Firstly, Luo & Wang (2024) has shown that the pure convolution structure has better efficiency than the costly attention-based structures in time series modeling, while maintaining comparable or better performance. Besides, contrastive discrepancy learning doesn't only work on attention. It can also work on convolution theoretically. For example, by using **multi-branch convolution structures**, we can also produce different representations of the same input and calculate their discrepancy for contrastive learning, which provides theoretical support for the fitness between pure convolution structures and contrastive-based methods. The idea of multi-branch convolution can be dated back to Inception (Szegedy et al., 2014) in Computer Vision (CV), where the different convolution branches can provide different representations under different kernel sizes. In the latest studies, the differences between each convolution branch are not only limited to the different kernel sizes, but also different dilation ratios, sparse ratios, and kernel shapes (Ding et al., 2022; 2023; Liu et al., 2022a). These CV methods treat different branches from the perspective of aggregation. They aggregate the representations from different branches to obtain a new and more informative representation. But in this paper, we take a novel and opposite perspective on the multi-branch convolution structure. **We turn to distinguish the representation discrepancy between each branch and further derive a more distinguishable anomaly criterion based on this discrepancy.**

In detail, we design a dual-branch convolution structure. One branch is of a large kernel dense convolution and the other branch is of a dilated convolution with the equivalent kernel size. And we provide a simple scenario in Figure 1 for illustration. In this case, we use the simplest dual-branch convolution structure, in which we only adopt a single depth-wise convolution layer in each branch and fix the convolution kernels as the simple mean filters. And we intent to detect the global point anomalies in this case. As shown in Figure 1, when given a time series with anomalies, the dense convolution can't skip the anomalies. But the dilated convolution can skip the anomalies and therefore can produce a different representation from that of the dense convolution. By contrast, since all normal points share the similar latent patterns (Xu et al., 2021; Yang et al., 2023), skipping some normal points doesn't affect the representation too much. When given a normal time series, two convolution branches can produce similar representations. As a result, the representation discrepancy between two convolution branches are more likely to be larger when faced with anomaly instances in this case, which can serve as a distinguishable criterion to detect the anomalies. Above discussion on Figure 1 shows that just a simplest and even non-trainable dual-branch convolution structure can have certain anomaly detection capability, verifying its great potential. And in our real implementation which will be introduced in Section 3, we adopt a deeper network structure to bring better representation capacity. And we also make the weights of convolution kernels trainable. Thus our model can learn to handle more types of anomalies and can be used in more difficult anomaly detection tasks.

Based on above motivations, we propose a dual-branch convolution structure called **AnomalyTCN** as a more efficient and effective solution for time series anomaly detection. In terms of the learning task, we adopt the idea of contrastive representation and discrepancy learning. Our model works purely based on the discrepancy of two branches in a contrastive manner, successfully getting rid of the classic but less-robust reconstruction task. In terms of the backbone design, we adopt a simple and light-weight pure convolution structure to avoid the complicated attention computation, therefore having much more advantages in efficiency. And in terms of the anomaly criterion, we adopt the

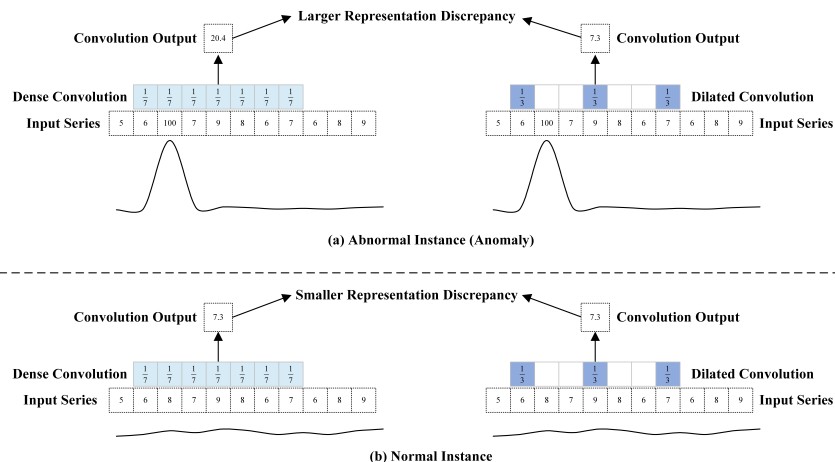

Figure 1: An illustration of using dual-branch convolution structure with contrastive representation for time series anomaly detection. The representation discrepancy between two convolution branches can be used as a more distinguishable criterion to detect the anomalies.

representation discrepancy between two convolution branches as a more distinguishable criterion, therefore leading to the excellent detection performance. Experimentally, our AnomalyTCN achieves the consistent state-of-the-art performance on various time series anomaly detection tasks. When competing comparably or superiorly than state-of-the-art methods, our method also shows greater efficiency by saving 83.6% running time and 20.1% memory usage. Our contributions are as follows:

- We successfully combine the pure convolution structure with contrastive discrepancy learning and novelly propose a dual-branch convolution structure for contrastive-based time series anomaly detection. And we further adopt the representation discrepancy between two branches as a more distinguishable criterion to bring better detection performance.

- Our AnomalyTCN achieves the consistent state-of-the-art anomaly detection results on various benchmarks, demonstrating the performance superiority. Extensive model analysis and insightful case studies are given.

- To solve the efficiency issue in previous detection methods, we propose a pure convolution structure as a more efficient solution. Our AnomalyTCN shows much greater efficiency than the state-of-the-art models, thus providing a better balance of performance and efficiency.

## 2 RELATED WORK

We briefly provide some related works as follows. Please refer to Appendix C for more related works.

### 2.1 CONTRASTIVE-BASED TIME SERIES ANOMALY DETECTION

The classic reconstruction-based methods suffer from the performance bottleneck for they are easier to be interfered by anomalies during the training phase (Yang et al., 2023). And the reconstruction error is also not good enough to be an anomaly criterion (Xu et al., 2021). To address these issues, some contrastive-based methods are proposed. Their insight is that, since normal points usually share the similar latent patterns, the representation discrepancy under different views for normal points are less than that for anomalies. Thus they adopt the representation discrepancy as a better anomaly detection criterion. For example, Anomaly Transformer (Xu et al., 2021) adopts the association discrepancy as a complementary to the reconstruction error, resulting in a hybrid anomaly criterion. DCdetecor (Yang et al., 2023) directly removes the reconstruction task. It proposes a dual-attention structure and adopts a contrastive representation task. Given an input, it calculates two attention maps from two different views and detects anomalies by the larger attention discrepancy. But these two methods bring severe computation costs due to the quadratic complexity in their attention computation. To improve efficiency, we design an attention-free contrastive-based method in this paper.

### 2.2 MULTI-BRANCH CONVOLUTION STRUCTURE

The idea of multi-branch convolution can be traced back to the early exploration in CV, where the family of Inception networks (Szegedy et al., 2014; 2015; 2016; Chollet, 2016; Liu et al., 2020)

proposes a multi-branch convolution structure with different kernel sizes to aggregate features under different receptive fields. In 2020s, with the proposal of structural re-parameterization (Ding et al., 2021), the idea of multi-branch convolution is revitalized. RepLKnet (Ding et al., 2022) proposes that a dual-branch convolution structure with a large kernel and a small kernel can effectively enlarge the receptive field. Following it, SLaK (Liu et al., 2022a) further proposes a tri-branch convolution structure consisting of two rectangular large kernel and a square small kernel. And UnirepLKnet (Ding et al., 2023) proposes a multi-branch convolution structure with five different dilation ratios and kernel sizes. These CV methods tend to aggregate the representation outputs from different branches to obtain a more informative new representation. But in this paper, we take a novel and opposite perspective on the multi-branch convolution structure. We distinguish the representation discrepancy between each branch and further use it as a distinguishable criterion for time series anomaly detection.

## 3 METHOD

Considering a length-$T$ multivariate time series with $M$ variates: $\mathbf{X} = \{x_1, x_2, \cdots, x_T\}$, where $x_t \in \mathbb{R}^M$ represents the observation at time $t$, the goal of time series anomaly detection is to determine whether $x_t$ is anomalous or not. To provide a better balance of performance and efficiency, we propose **AnomalyTCN** as a more efficient and effective solution. Our AnomalyTCN adopts a dual-branch convolution structure and works purely based on the representation discrepancy of two branches in a contrastive manner, maintaining the performance superiority of contrastive-based methods. Meanwhile, since we adopt a simple and light-weight pure convolution structure to avoid the complicated attention computation, our method can enjoy much more advantages in efficiency.

As shown in Figure 2 (a), we adopt a contrastive framework free of negative samples. And different from CV, we use no data argumentation and thus the inputs for two branches are totally the same. Despite the exact same input, the two branches can still produce different representations from different views thanks to our asymmetric designs. One is the asymmetric structure designs adopted in two branches and the other is the asymmetric training designs introduced by stop-gradient operation. In terms of structure designs, one branch is from the view of dense convolution, which will cover all time points. The other branch is from the view of dilated convolution, which can skip some time points. As a result, these two asymmetric convolution branches can produce different representations of the same input from different views and further calculate their representation discrepancy for contrastive learning (Section 3.1). As illustrated in Figure 1, this representation discrepancy between two convolution branches can be derived as a more distinguishable anomaly criterion in testing phase, which brings better detection performance (Section 3.3). And in training phase, we train our model purely based on discrepancy learning in a contrastive manner and introduce asymmetry by stop-gradient to ensure the training process is non-trivial (Section 3.2).

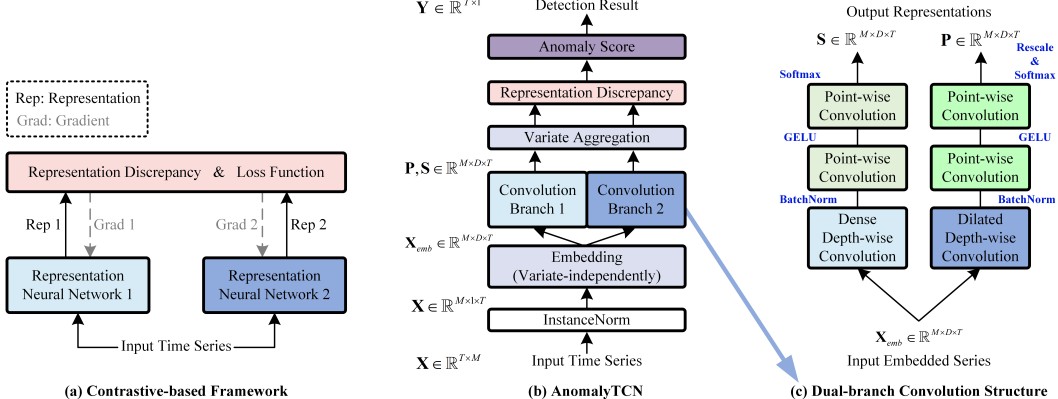

Figure 2: Workflow of AnomalyTCN. (a) The framework for contrastive discrepancy learning free of negative samples. (b) The structure of AnomalyTCN. (c) The dual-branch convolution structure.

### 3.1 STRUCTURE OF ANOMALYTCN

The overall structure of AnomalyTCN is shown in Figure 2 (b) and (c). The forward process in AnomalyTCN mainly includes the input embedding module and the dual-branch convolution structure.

**Input Embedding Module**  As with DCdetector (2023), we adopt the idea of variate-independence. Given $\mathbf{X} \in \mathbb{R}^{T \times M}$ as a multivariate time series input with $M$ variates, we embed and calculate the representations for each of the $M$ variates independently, and then aggregate them by mean pooling at the end of the forward process. And we also adopt the instance normalization (Kim et al., 2021), a widely-used technique in various latest time series models, to make the training process more stable. In detail, we first normalize the input time series $\mathbf{X}$ by InstanceNorm. Then after permuting and unsqueezing the shape into $\mathbf{X} \in \mathbb{R}^{M \times 1 \times T}$, we embed the time series variate-independently into $D$-dimensional embedded series: $\mathbf{X}_{emb} = \text{Embedding}(\mathbf{X})$, where $\mathbf{X}_{emb} \in \mathbb{R}^{M \times D \times T}$ is the input embedding and $T$, $M$ and $D$ are the sizes of temporal, variate and feature dimensions. Embedding is implemented by point-wise convolution stem layer (Luo & Wang, 2024). Then $\mathbf{X}_{emb}$ will be passed into the dual-branch convolution structure to obtain the representations from two different views.

**Dual-branch Convolution Structure**  Our dual-branch structure adopts asymmetric structure designs to produce different representations of the same input. And to provide a better representation capacity, each branch is designed as a modern convolution block (Liu et al., 2022b; Luo & Wang, 2024), which includes a depth-wise large kernel convolution layer to mix the temporal information and two successive point-wise convolution layers to mix the feature information. And a GELU activation (Hendrycks & Gimpel, 2016) is adopted between two point-wise convolution layers to provide nonlinearity. Besides, to provide sufficient structural asymmetry, the depth-wise convolution layers in two branches adopt different structure settings, where one is a dense convolution and the other is a dilated convolution. As mentioned in Figure 1, this structure difference can bring certain anomaly detection capability even without training, which ensures the performance. Meanwhile, the successive point-wise convolution modules in two branches do not share the weights, even though they have the same structure settings. The forward process in dual-branch structure is formalized as:

$$\mathbf{S} = \text{Softmax}(\text{Conv}_1(\mathbf{X}_{emb}))$$
$$\mathbf{P} = \text{Softmax}(\text{Conv}_2(\mathbf{X}_{emb})) \tag{1}$$

$\text{Conv}_1(\cdot)$ and $\text{Conv}_2(\cdot)$ are the dense convolution branch and dilated convolution branch, respectively. And $\text{Softmax}(\cdot)$ normalizes the representations along the temporal dimension. Following DCdetector (2023), we further rescale $\mathbf{P}$ by dividing the row sum along the feature dimension. Since we only rescale $\mathbf{P}$ and maintain $\mathbf{S}$, this operation can further provide asymmetry. Afterwards, these two output representations $\mathbf{P}$ and $\mathbf{S}$ will be used for discrepancy learning during training phase (Section 3.2) and can be derived as a distinguishable anomaly criterion in testing phase (Section 3.3).

## 3.2 TRAINING PHASE: DISCREPANCY LEARNING

**Loss Function**  After obtaining the two representations $\mathbf{P}$ and $\mathbf{S}$ from two convolution branches, we formalize a loss function based on Symmetrized Kullback–Leibler divergence (KL divergence) to measure the discrepancy of two representations. And the loss function for $\mathbf{P}$ and $\mathbf{S}$ are defined as:

$$\mathcal{L}_{\mathbf{P}}(\mathbf{P}, \mathbf{S}; \mathbf{X}) = \text{KL}(\mathbf{P}, \text{Stopgrad}(\mathbf{S})) + \text{KL}(\text{Stopgrad}(\mathbf{S}), \mathbf{P}) \tag{2}$$
$$\mathcal{L}_{\mathbf{S}}(\mathbf{P}, \mathbf{S}; \mathbf{X}) = \text{KL}(\mathbf{S}, \text{Stopgrad}(\mathbf{P})) + \text{KL}(\text{Stopgrad}(\mathbf{P}), \mathbf{S}) \tag{3}$$

where $\mathbf{X} \in \mathbb{R}^{T \times M}$ is the input time series, $\text{KL}(\cdot||\cdot)$ is the KL divergence, $\mathbf{P}, \mathbf{S} \in \mathbb{R}^{M \times D \times T}$ are the representation results from the two convolution branches. Stopgrad is the stop-gradient operation, which is used in our loss function to train two branches asymmetrically. Then similar to DCdetector (2023), the total loss function $\mathcal{L}$ for input time series $\mathbf{X}$ is defined as:

$$\mathcal{L} = \mathcal{L}_{\mathbf{P}} - \mathcal{L}_{\mathbf{S}} \tag{4}$$

Compared with previous reconstruction-based models which are easily interfered by anomalies in the training process, our AnomalyTCN is trained purely based on the representation discrepancy in a contrastive manner (Equation 2 3 4). Since not utilizing any reconstruction parts, our method can reduce the disruption from anomalies in the training phase and leads to performance improvement.

**Stop-gradient Operation**  The stop-gradient operation (Stopgrad) can introduce asymmetry into the training process, which helps to avoid trivial solutions. When using Stopgrad, the branch $\text{Conv}_1$ is only supervised by $\mathcal{L}_{\mathbf{S}}$ and the other branch $\text{Conv}_2$ is only supervised by $\mathcal{L}_{\mathbf{P}}$. This asymmetric supervisions can bring better training results (Yang et al., 2023; Chen et al., 2020a) and our AnomalyTCN indeed gains the best performance when equipped with Stopgrad. But we are also surprised to find that AnomalyTCN still works well and outperforms many baselines even if not using Stopgrad, which is different from the contrastive-based methods in CV that the lack of Stopgrad will lead to a trivial solution. And we will discuss this finding in detail in Section 5.3.

### 3.3 Testing Phase: Anomaly Criterion

We adopt the same anomaly score as in DCdetector (2023), which is a universal point-wise anomaly score for the dual-branch structure based on its representation discrepancy. Given the representations $\mathbf{P}$ and $\mathbf{S}$ from two convolution branches, the anomaly score of $\mathbf{X} \in \mathbb{R}^{T \times M}$ is:

$$\text{AnomalyScore}(\mathbf{X}) = \text{Softmax}\left(-(\text{KL}(\mathbf{P}, \mathbf{S}) + \text{KL}(\mathbf{S}, \mathbf{P}))\right) \tag{5}$$

Based on this point-wise anomaly score where anomalies will lead to higher scores than normal points, a hyperparameter threshold $\delta$ is used to decide whether a time point is an anomaly (1) or not (0). If the score exceeds the threshold, the output $\mathbf{Y} \in \mathbb{R}^{T \times 1}$ is an anomaly:

$$\mathbf{Y}_t = \begin{cases} 1: \text{anomaly} & \text{AnomalyScore}(\mathbf{X}_t) \geq \delta \\ 0: \text{normal} & \text{AnomalyScore}(\mathbf{X}_t) < \delta. \end{cases} \tag{6}$$

where the index $_t$ means the $t$-th points in time series $\mathbf{X}$ and the range of $t$ is $[0, T-1]$.

## 4 Experiments

We extensively evaluate our AnomalyTCN on seven benchmarks from five real-world applications.

**Setups** We conduct experiments on following datasets, including five real-world datasets and two NeurIPS-TS datasets: (1) SMD (Server Machine Dataset) Su et al. (2019), (2) PSM (Pooled Server Metrics) Abdulaal et al. (2021), (3) MSL (Mars Science Laboratory rover) (Hundman et al., 2018), (4) SMAP (Soil Moisture Active Passive satellite) (Hundman et al., 2018), (5) SWaT (Secure Water Treatment) Mathur & Tippenhauer (2016), (6) NeurIPS-TS-SWAN (Angryk et al., 2020; Lai et al., 2021), (7) NeurIPS-TS-GECCO Rehbach et al. (2018); Lai et al. (2021). Each dataset contains one continuous long time series, and we obtain input samples from the continuous long time series with a fixed length sliding window. For real-world benchmarks, we follow the well-established protocol in Shen et al. (2020); Xu et al. (2021) to set the window length as 100 for all datasets. For NeurIPS-TS benchmarks, we follow Lai et al. (2021); Yang et al. (2023) to set the window length as 36 for NeurIPS-TS-SWAN and 90 for NeurIPS-TS-GECCO. More details of datasets and implementation details are summarized in Appendix A and B.

**Baselines** We comprehensively compare our model with various strong baselines, including (1) the classic reconstruction-based models: D3R (2023), InterFusion (2021), BeatGAN (2019), OmniAnomaly (2019), LSTM-VAE (2018); (2) the autoregression-based models: CL-MPPCA (2019), LSTM (2018), VAR (1976); (3) the density-estimation models: DAGMM (2018), MPPCACD (2017), LOF (2000); (4) the clustering-based methods: ITAD (2020), THOC (2020), Deep-SVDD (2018); (5) the classic machine learning methods: IsolationForest (2008), OCSVM (2004); (6) the change point detection and time series segmentation methods: TS-CP2 (2021), U-Time (2019), BOCPD (2007). (7) the advanced reconstruction-based method with general time series backbones: aLLM4TS (2024), ModernTCN (2024), GPT4TS (2023), TimesNet (2023); (8) the pure contrastive-based method: DCdetor (2023) and (9) the hybrid contrastive-based and reconstruction-based method: Anomaly Transformer (2021). Specially for NeurIPS-TS benchmarks, some baselines in the original paper (Lai et al., 2021) are also included, i.e., AutoEncoder (2014), Autoregression (2005), LSTM-RNN (2016), OCSVM-based subsequence clustering (OCSVM*) (2004), IForest-based subsequence clustering (IForest*) (2008), Gradient boosting regression (GBRT) (2021) and Matrix Profile (2016).

### 4.1 Main Results

**Real-world datasets** We extensively evaluate our model on five real-world datasets with more than 20 competitive baselines. The results are shown in Table 1. Overall, our AnomalyTCN achieves the consistent state-of-the-art performance with the highest average F1-score. In detail, it performs the best in most cases and outperforms other baselines by a large magrin, which verifies the effectiveness of our method. Meanwhile, the other pure contrastive-based method DCdetor (2023) also achieves the second best performance, further verifying the performance superiority of contrastive-based methods than the classic reconstruction-based ones. Besides, although without the complicated attention computation, our AnomalyTCN still performs excellently and surpasses other attention-based contrastive methods (i.e., DCdetor (2023) and Anomaly Transformer (2021)). This comparison proves our claim that contrastive discrepancy learning doesn't only work on attention, thus verifying the soundness of our solution to design a more efficient contrastive-based method by replacing the costly attention mechanism with the light-weight and simple pure convolution structure.

Table 1: Results of anomaly detection in real-world datasets. The P, R and F1 represent the precision, recall and F1-score (%). F1-score is the harmonic mean of precision and recall. A higher value of P, R and F1 indicates a better performance. The best results are in **bold** and the second are underlined. The models are ranked from lowest to highest based on the average F1-score of the five datasets.

| Datasets | SMD | | | MSL | | | SMAP | | | SWaT | | | PSM | | | Avg F1 |
|---|---|---|---|---|---|---|---|---|---|---|---|---|---|---|---|---|
| Metrics | P | R | F1 | P | R | F1 | P | R | F1 | P | R | F1 | P | R | F1 | (%) |
| OCSVM | 44.34 | 76.72 | 56.19 | 59.78 | 86.87 | 70.82 | 53.85 | 59.07 | 56.34 | 45.39 | 49.22 | 47.23 | 62.75 | 80.89 | 70.67 | 60.25 |
| LOF | 56.34 | 39.86 | 46.68 | 47.72 | 85.25 | 61.18 | 58.93 | 56.33 | 57.60 | 72.15 | 65.43 | 68.62 | 57.89 | 90.49 | 70.61 | 60.94 |
| IForest | 42.31 | 73.29 | 53.64 | 53.94 | 86.54 | 66.45 | 52.39 | 59.07 | 55.53 | 49.29 | 44.95 | 47.02 | 76.09 | 92.45 | 83.48 | 61.22 |
| U-Time | 65.95 | 74.75 | 70.07 | 57.20 | 71.66 | 63.62 | 49.71 | 56.18 | 52.75 | 46.20 | 87.94 | 60.58 | 82.85 | 79.34 | 81.06 | 65.62 |
| DAGMM | 67.30 | 49.89 | 57.30 | 89.60 | 63.93 | 74.62 | 86.45 | 56.73 | 68.51 | 89.92 | 57.84 | 70.40 | 93.49 | 70.03 | 80.08 | 70.18 |
| ITAD | 86.22 | 73.71 | 79.48 | 69.44 | 84.09 | 76.07 | 82.42 | 66.89 | 73.85 | 63.13 | 52.08 | 57.08 | 72.80 | 64.02 | 68.13 | 70.92 |
| VAR | 78.35 | 70.26 | 74.08 | 74.68 | 81.42 | 77.90 | 81.38 | 53.88 | 64.83 | 81.59 | 60.29 | 69.34 | 90.71 | 83.82 | 87.13 | 74.66 |
| MMPCACD | 71.20 | 79.28 | 75.02 | 81.42 | 61.31 | 69.95 | 88.61 | 75.84 | 81.73 | 82.52 | 68.29 | 74.73 | 76.26 | 78.35 | 77.29 | 75.74 |
| CL-MPPCA | 82.36 | 76.07 | 79.09 | 73.71 | 88.54 | 80.44 | 86.13 | 63.16 | 72.88 | 76.78 | 81.50 | 79.07 | 56.02 | 99.93 | 71.80 | 76.66 |
| TS-CP2 | 87.42 | 66.25 | 75.38 | 86.45 | 68.48 | 76.42 | 87.65 | 83.18 | 85.36 | 81.23 | 74.10 | 77.50 | 82.67 | 78.16 | 80.35 | 79.00 |
| BeatGAN | 72.90 | 84.09 | 78.10 | 89.75 | 85.42 | 87.53 | 92.38 | 55.85 | 69.61 | 64.01 | 87.46 | 73.92 | 90.30 | 93.84 | 92.04 | 80.24 |
| BOCPD | 70.90 | 82.04 | 76.07 | 80.32 | 87.20 | 83.62 | 84.65 | 85.85 | 85.24 | 89.46 | 70.75 | 79.01 | 80.22 | 75.33 | 77.70 | 80.33 |
| Deep-SVDD | 78.54 | 79.67 | 79.10 | 91.92 | 76.63 | 83.58 | 89.93 | 56.02 | 69.04 | 80.42 | 84.45 | 82.39 | 95.41 | 86.49 | 90.73 | 80.97 |
| LSTM-VAE | 75.76 | 90.08 | 82.30 | 85.49 | 79.94 | 82.62 | 92.20 | 67.75 | 78.10 | 76.00 | 89.50 | 82.20 | 73.62 | 89.92 | 80.96 | 81.24 |
| LSTM | 78.55 | 85.28 | 81.78 | 85.45 | 82.50 | 83.95 | 89.41 | 78.13 | 83.39 | 86.15 | 83.27 | 84.69 | 76.93 | 89.64 | 82.80 | 83.32 |
| OmniAnomaly | 83.68 | 86.82 | 85.22 | 89.02 | 86.37 | 87.67 | 92.49 | 81.99 | 86.92 | 81.42 | 84.30 | 82.83 | 88.39 | 74.46 | 80.83 | 84.69 |
| InterFusion | 87.02 | 85.43 | 86.22 | 81.28 | 92.70 | 86.62 | 89.77 | 88.52 | 89.14 | 80.59 | 85.58 | 83.01 | 83.61 | 83.45 | 83.52 | 85.70 |
| TimesNet | 88.66 | 83.14 | 85.81 | 83.92 | 86.42 | 85.15 | 92.52 | 58.29 | 71.52 | 86.76 | 97.32 | 91.74 | 98.19 | 96.76 | 97.47 | 86.34 |
| ModernTCN | 87.86 | 83.85 | 85.81 | 83.94 | 85.93 | 84.92 | 93.17 | 57.69 | 71.26 | 91.83 | 95.98 | 93.86 | 98.09 | 96.38 | 97.23 | 86.62 |
| GPT4TS | 88.89 | 84.98 | 86.89 | 82.00 | 82.91 | 82.45 | 90.60 | 60.95 | 72.88 | 92.20 | 96.34 | 94.23 | 98.62 | 95.68 | 97.13 | 86.72 |
| aLLM4TS | 87.87 | 83.09 | 85.42 | 81.58 | 82.95 | 82.26 | 85.40 | 71.84 | 78.04 | 97.90 | 92.53 | 94.57 | 98.47 | 95.94 | 97.19 | 87.51 |
| THOC | 79.76 | 90.95 | 84.99 | 88.45 | 90.97 | 89.69 | 92.06 | 89.34 | 90.68 | 83.94 | 86.36 | 85.13 | 88.14 | 90.99 | 89.54 | 88.01 |
| D3R | 69.41 | 98.79 | 81.54 | 87.71 | 81.20 | 84.33 | 93.88 | 92.10 | 92.99 | 89.33 | 95.81 | 92.46 | 98.43 | 96.00 | 97.20 | 89.70 |
| Anomaly Transformer | 87.21 | 93.77 | **90.37** | 92.24 | 96.42 | 94.28 | 93.66 | 99.28 | 96.39 | 88.75 | 98.37 | 93.31 | 96.62 | 98.27 | 97.44 | 94.36 |
| DCdetector | 88.42 | 88.14 | 88.28 | 92.50 | 97.36 | 94.87 | 94.33 | 98.70 | 96.47 | 93.10 | 99.96 | 96.41 | 96.93 | 97.79 | 97.36 | 94.68 |
| AnomalyTCN (Ours) | 86.88 | 92.93 | 89.80 | 92.76 | 99.71 | **96.11** | 93.67 | 99.58 | **96.53** | 93.33 | 100.00 | **96.55** | 97.08 | 98.83 | **97.95** | **95.39** |

**NeurIPS-TS benchmark** NeurIPS-TS-SWAN and NeurIPS-TS-GECCO are more challenging than above real-world datasets. These two datasets are with more types of anomalies and have the highest and the lowest anomaly ratios (32.6% in NeurIPS-TS-SWAN and 1.1% in NeurIPS-TS-GECCO). To fully evaluate the performance of AnomalyTCN, we also conduct experiments on these two challenging datasets and chose the models that perform well in the real-world datasets as strong baselines. As shown in Figure 3, our AnomalyTCN still achieves the consistent state-of-the-art performance. Specifically, it completely outperforms other methods and especially gains at least 15% promotion in NeurIPS-TS-GECCO, verifying the effectiveness of our model on various anomalies.

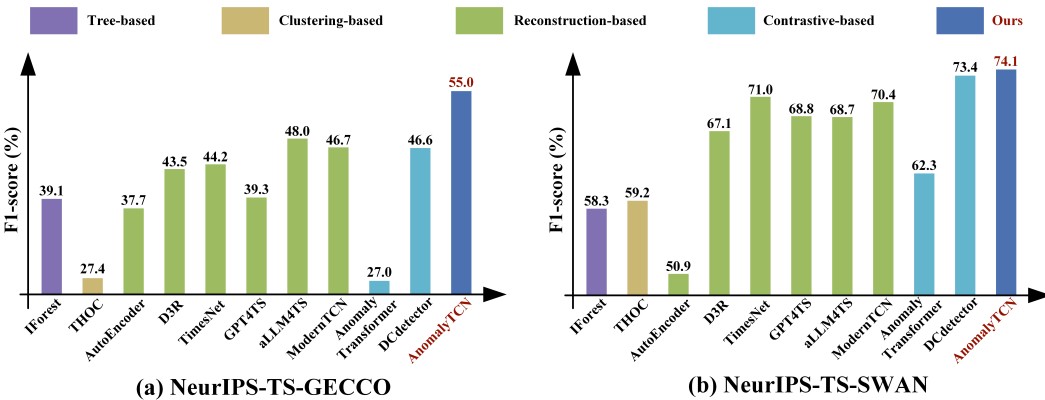

Figure 3: Results of anomaly detection in NeurIPS-TS benchmarks. A higher F1-score indicates better performance. See Table 15 in Appendix H for full results with more baselines.

**Showcases Analysis** We provide the showcases under different types of anomalies as an intuitive comparision. Please refer to Appendix D for details.

## 4.2 MODEL EFFICIENCY

We comprehensively compare the performance, running time, and memory usage of our AnomalyTCN with the previous state-of-the-art model DCdetector (2023). As shown in Table 2, while achieving the consistent state-of-the-art performance on various time series anomaly detection tasks, our AnomalyTCN can save 83.6% running time and 20.1% memory usage, therefore providing a better balance of performance and efficiency than other competitors. Particularly in terms of running time, our AnomalyTCN surpasses DCdetector (2023) by a large margin, demonstrating its great potential for real-time anomaly detection applications. And the efficiency of AnomalyTCN can be further improved with the help of a more light-weight variate-mixing embedding method, which is discussed in Appendix G.

Table 2: Model efficiency comparison of AnomalyTCN and DCdetector. *F1*, *Time* and *Mem* means the F1-score (%), Averaged running time of 100 iterations (s/iter) and Peak memory usage (GB). A higher *F1* means better performance and a lower *Time* and *Mem* indicate better efficiency. The best ones are in **bold**.

| Dataset | SMD | | | MSL | | | SMAP | | | SWaT | | | PSM | | | NIPS-TS-SWAN | | | NIPS-TS-GECCO | | |
|---|---|---|---|---|---|---|---|---|---|---|---|---|---|---|---|---|---|---|---|---|---|
| Model | F1 | Time | Mem | F1 | Time | Mem | F1 | Time | Mem | F1 | Time | Mem | F1 | Time | Mem | F1 | Time | Mem | F1 | Time | Mem |
| DCdetector | 88.28 | 0.062 | 18.8 | 94.87 | 0.102 | 25.9 | 96.47 | 0.071 | 17.5 | 96.41 | 0.130 | 24.7 | 97.36 | 0.104 | 38.3 | 73.4 | 0.046 | 10.9 | 46.6 | 0.052 | 10.4 |
| AnomalyTCN (Ours) | **89.80** | **0.015** | **17.5** | **96.11** | **0.020** | **24.4** | **96.53** | **0.010** | **14.3** | **96.55** | **0.021** | **22.8** | **97.95** | **0.010** | **14.3** | **74.1** | **0.008** | **8.9** | **55.0** | **0.007** | **8.2** |
| Promotion | 1.7% | 75.8% | 6.9% | 1.3% | 80.4% | 5.8% | 0.1% | 85.9% | 18.3% | 0.1% | 83.8% | 7.7% | 0.6% | 90.4% | 62.7% | 1.0% | 82.6% | 18.3% | 18.0% | 86.5% | 21.2% |

## 5 MODEL ANALYSIS

We study the key elements of our dual-branch convolution structure from following aspects. In Section 5.1, since the idea of asymmetric structure designs plays a vital role in our dual-branch structure, we conduct ablation on it to highlight its importance and to verify the effectiveness of our two asymmetric structure designs. Then in Section 5.2, we will further discuss how to specifically design an effective dual-branch structure by studying the impact of different settings in our method. In Section 5.3, we also discuss how to effectively train a dual-branch structure by studying the impact of stop-gradient on the training process. And more studies on our model are provided in Appendix F.

### 5.1 ABLATION STUDY

In this section, we conduct ablation to highlight the importance of using structural asymmetry in dual-branch structures and verify the effectiveness of our two asymmetric structure designs.

Our ablation starts from the case of *use same structure settings with weight-sharing in two branches*, which is a structure design without any structural asymmetry. As shown in Table 3, this design provides a trivial solution and totally fails in anomaly detection tasks. The reason is as follow. Due to the exact same input $\mathbf{X}$, the output representations $\mathbf{P}$ and $\mathbf{S}$ are always the same when there is no asymmetry in structure designs, regardless the input is normal or abnormal. Thus the anomaly score, which is calculated based on the representation discrepancy between $\mathbf{P}$ and $\mathbf{S}$, is always 0 in this case and can not be used to distinguish the anomalies, making it fail in anomaly detection tasks. This result demonstrates the necessity of using structural asymmetry in dual-branch structures.

To provide sufficient structural asymmetry, we introduce two asymmetric strucutre designs in this paper. Firstly, we add the additional rescale operation in one of the branches to bring asymmetry, making the model work for anomaly detection. Secondly, we adopt different structure settings in two branches to bring fully structural asymmetry. As shown from the top to the bottom of Table 3, we enlarge the structure difference between two branches step by step, in which we *remove the weight-sharing* and further change to *use different structure settings in two branches*. During this process, we observe continuous performance improvement with the increasement of structure difference. And our method, which equips with both of our two asymmetric structure designs, achieves the best performance. These results verify the importance of bringing more structural asymmetry in dual-branch structures and prove the effectiveness of our two asymmetric structure designs.

Meanwhile, we also compare our additional rescale operation with the additional predictor layer, which is a common structure design adopted by the contrastive-based methods in CV to provide asymmetry (He et al., 2019; Chen & He, 2020). This additional predictor layer is implemented by

multi-layer perceptron (MLP) and is also added in one of the branches. As shown in Table 3, our rescale operation can perform better in this comparison, which further validates its effectiveness.

Table 3: Study on asymmetric structure designs. All desings are trained with $\mathrm{Stopgrad}$. A higher P, R, F1 (%) means better performance, and the best ones are in **bold**.

| Dataset | MSL | | | SMAP | | | SWaT | | | PSM | | |
|---|---|---|---|---|---|---|---|---|---|---|---|---|
| Metric | P | R | F1 | P | R | F1 | P | R | F1 | P | R | F1 |
| Ablation on asymmetric structure designs | | | | | | | | | | | | |
| Starting point: Use same structure settings with weight-sharing in two branches | 0.00 | 0.00 | 0.00 | 0.00 | 0.00 | 0.00 | 0.00 | 0.00 | 0.00 | 0.00 | 0.00 | 0.00 |
| Add the additional rescale operation in one branch | 76.60 | 60.93 | 67.87 | 89.79 | 32.93 | 48.19 | 93.38 | 96.79 | 95.05 | 97.46 | 94.69 | 96.06 |
| Change to use same structure settings in two branches but remove weight-sharing | 92.26 | 93.12 | 92.68 | 94.35 | 91.39 | 92.85 | 93.29 | 97.87 | 95.52 | 97.01 | 95.86 | 96.43 |
| **Change to use different structure settings in two branches (Ours)** | 92.76 | 99.71 | **96.11** | 93.67 | 99.58 | **96.53** | 93.33 | 100.00 | **96.55** | 97.08 | 98.83 | **97.95** |
| Compared with CV method | | | | | | | | | | | | |
| Add the additional rescale operation in one branch (Our method) | 76.60 | 60.93 | 67.87 | 89.79 | 32.93 | 48.19 | 93.38 | 96.79 | 95.05 | 97.46 | 94.69 | 96.06 |
| Add the additional predictor layer in one branch (CV method) | 78.70 | 48.30 | 59.86 | 83.95 | 20.53 | 32.99 | 91.68 | 88.78 | 90.21 | 98.32 | 84.73 | 91.02 |

## 5.2 IMPACT OF DIFFERENT SETTINGS IN DUAL-BRANCH STRUCTURE

In main experiments, we adopt the combination of a dense convolution branch and a dilated convolution branch with equivalent kernel size to provide structural asymmetry. And we set the default kernel size as 7. In this section, we study the impact of different kernel sizes (receptive fields). And we also compare our method with another dual-branch convolution design that adopts the combination of two dense convolution branches with different kernel sizes. The results are shown in Table 4 and some discussions are as follows:

**(1)** Our method is robust to the choice of kernel sizes. It can work well with the common values of kernel size in modern convolution blocks, such as 9, 7 and 5 (Liu et al., 2022b).

**(2)** Although using a dilated convolution branch helps to bring sufficient structural asymmetry, the value of dilation ratio can not be too large in some datasets (e.g., SWaT). A possible reason is as follow. As illustrated in Figure 1, the dilated depth-wise convolution can help to detect the anomalies because it can skip some time points and thus produce a different representation from the dense one. Generally, skipping the abnormal points leads to a greater representation discrepancy than skipping the normal points, thus helping to detect the anomalies. But when we use a very large dilation ratio (e.g., $R = 4$), the dilated depth-wise convolution will skip a very large number of normal points. In some datasets, this behavior may cause a great representation discrepancy even under normal samples, thereby misjudging normal samples as abnormal and leading to performance degradation.

**(3)** The combination of two dense convolutions with different kernel sizes can help to detect the anomalies. But it is less robust and may fail in some datasets. This dual-branch design can detect the anomalies because the receptive fields of two branches are different, which brings structural asymmetry in two branches. As a result, these two branches can produce different representations of the same input from two different views and further calculate their representation discrepancy as a criterion for contrastive-based time series anomaly detection. But this kind of structural asymmetry also leads to an information loss in the small kernel branch because it can only see a smaller area than the large kernel branch. This information loss makes the small kernel branch not an appropriate view to produce contrastive representation for the same input, hence reducing the effectiveness of their representation discrepancy as an anomaly criterion and making it fail to accurately detect the anomalies in some cases. By contrast, in our method where the receptive fields of two branches are equivalent, it can avoid this information loss and achieve consistent better performance in all cases.

Above results validate the robustness of our dual-branch designs and highlight the effectiveness of using our combination of a dense convolution and a dilated convolution with equivalent kernel size.

## 5.3 STUDY ON STOP-GRADIENT OPERATION

In this section, we study the impact of stop-gradient operation ($\mathrm{Stopgrad}$). As shown in Table 5, AnomalyTCN obtains the best performance when both of the two $\mathrm{Stopgrad}$ are used, which validates the effectiveness of our asymmetric supervisions introduced by $\mathrm{Stopgrad}$. Surprisingly, if not using $\mathrm{Stopgrad}$, AnomalyTCN does not fall into a trivial solution and still outperforms many baselines, despite there are some performance degradation. A possible explanation is as follow.

Table 4: Impact of different settings in dual-branch structure. The settings are summarized in $[K_1, R_1, K_2, R_2]$, where $K$ and $R$ are the kernel size and dilation ratio for the convolution branch.

| Dataset | MSL | | | SMAP | | | SWaT | | | PSM | | |
|---|---|---|---|---|---|---|---|---|---|---|---|---|
| Metric | P | R | F1 | P | R | F1 | P | R | F1 | P | R | F1 |
| Combination of a dense and a dilated convolution branches with equivalent kernel size 9 | | | | | | | | | | | | |
| $[9,1,5,2]$ | 92.63 | 98.92 | 95.67 | 93.67 | 99.58 | 96.53 | 93.33 | 100.00 | 96.55 | 97.06 | 98.50 | 97.77 |
| $[9,1,3,4]$ | 92.63 | 98.92 | 95.67 | 93.66 | 99.46 | 96.47 | 89.90 | 81.76 | 85.64 | 96.98 | 97.30 | 97.14 |
| Combination of a dense and a dilated convolution branches with equivalent kernel size 7 | | | | | | | | | | | | |
| $[7,1,3,3]$ | 92.76 | 99.71 | 96.11 | 93.67 | 99.58 | 96.53 | 93.33 | 100.00 | 96.55 | 97.08 | 98.83 | 97.95 |
| Combination of a dense and a dilated convolution branches with equivalent kernel size 5 | | | | | | | | | | | | |
| $[5,1,3,2]$ | 92.71 | 99.69 | 96.07 | 93.67 | 99.58 | 96.53 | 93.33 | 100.00 | 96.55 | 97.11 | 98.66 | 97.87 |
| Combination of two dense convolution branches with different kernel sizes | | | | | | | | | | | | |
| $[7,1,9,1]$ | 92.62 | 98.76 | 95.59 | 93.55 | 89.07 | 91.25 | 92.05 | 87.11 | 89.51 | 97.07 | 98.23 | 97.65 |
| $[7,1,5,1]$ | 92.64 | 98.92 | 95.68 | 93.71 | 90.02 | 91.83 | 91.20 | 85.75 | 88.39 | 97.08 | 98.16 | 97.62 |
| $[7,1,3,1]$ | 92.63 | 98.92 | 95.67 | 93.66 | 90.98 | 92.30 | 90.67 | 85.04 | 87.76 | 97.04 | 97.92 | 97.48 |

Removing Stopgrad will affect the quality of training process and lead to performance degradation. But the structure difference in two asymmetric branches makes sure that AnomalyTCN can have certain anomaly detection capability even without training (as illustrated in Figure 1). Therefore our AnomalyTCN can still detect the anomalies after the low-quality training process caused by the removal of Stopgrad and ensure the performance superiority than many baselines. Meanwhile, the still competitive performance under the removal of Stopgrad also verifies that contrastive-based methods are less likely to collapse in time series anomaly detection, which can ensure the performance stability of contrastive-based time series anomaly detection frameworks and encourage further exploration in this direction. Please refer to Appendix C.4 for more our discussions.

Table 5: Study on Stopgrad. ✔ means adopting Stopgrad in the selected branch and ✘ means not adopting Stopgrad in the selected branch. A higher P, R, F1 (%) means better performance, and the best ones are in **bold**.

| Stopgrad | | MSL | | | SMAP | | | SWaT | | | PSM | | |
|---|---|---|---|---|---|---|---|---|---|---|---|---|---|
| Dense Conv Branch | Dilated Conv Branch | P | R | F1 | P | R | F1 | P | R | F1 | P | R | F1 |
| ✘ | ✘ | 90.79 | 86.66 | 88.68 | 92.63 | 93.82 | 93.22 | 93.22 | 95.59 | 94.39 | 97.25 | 93.85 | 95.52 |
| ✔ | ✘ | 92.57 | 96.70 | 94.59 | 93.65 | 99.14 | 96.31 | 93.50 | 99.57 | 96.44 | 96.99 | 97.96 | 97.48 |
| ✘ | ✔ | 91.75 | 87.95 | 89.81 | 91.81 | 96.34 | 94.02 | 93.25 | 100.00 | 96.51 | 97.60 | 97.62 | 97.61 |
| ✔ | ✔ | 92.76 | 99.71 | **96.11** | 93.67 | 99.58 | **96.53** | 93.33 | 100.00 | **96.55** | 97.08 | 98.83 | **97.95** |

## 6 CONCLUSION AND FUTURE WORK

This paper focuses on the rising contrastive-based method for time series anomaly detection. To solve the efficiency issue in previous costly attention-based solutions, we propose **AnomalyTCN** as a more efficient and effective solution. Technically, a dual-branch convolution structure is proposed to produce different representations of the same input from two different views. Then we calculate the representation discrepancy of these two convolution branches for contrastive learning and further derive a more distinguishable anomaly criterion based on this discrepancy, resulting in better detection performance. Besides, since adopting a simple and light-weight pure convolution structure to avoid the complicated attention computation, our method also shows much better efficiency superiority. Experimentally, extensive results validate that our AnomalyTCN is an ideal solution for time series anomaly detection with a better balance of performance and efficiency. Meanwhile, our study also reveals the possibility of combining contrastive-based anomaly detection frameworks with other efficient time series backbones beyond the costly attention mechanism, demonstrating the great potential in this novel research direction.

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

## A  DATASET

**Datasets**  Here is a description of the datasets: (1) SMD (Server Machine Dataset) is a 5-week-long compute cluster dataset that is collected from a Internet company with 38 dimensions (Su et al., 2019). (2) PSM (Pooled Server Metrics) is collected from multiple eBay server nodes with 26 dimensions (Abdulaal et al., 2021). (3) MSL (Mars Science Laboratory rover) and SMAP (Soil Moisture Active Passive satellite) are collected by NASA. MSL contains the sensor anomaly data of spacecraft monitoring systems with 55 dimensions and SMAP contains the telemetry anomaly data with 25 dimensions (Hundman et al., 2018). (4) SWaT (Secure Water Treatment) is collected from the critical infrastructure system under continuous operations with 51 dimensions (Mathur & Tippenhauer, 2016). (5) NeurIPS-TS-SWAN and NeurIPS-TS-GECCO are introduced by Lai et al. (2021) and include more types of time series anomalies. NeurIPS-TS-SWAN (Angryk et al., 2020) is extracted from solar photospheric vector magnetograms in Spaceweather HMI Active Region Patch series and NeurIPS-TS-GECCO (Rehbach et al., 2018) is a drinking water quality dataset for the Internet of Things. And we summarize the statistical details of these seven adopted benchmark datasets in Table 6. These datasets include various types of multivariate time series scenarios with different anomaly ratios. The training and validation subsets are splited from the unlabeled data with a spilt ratio of 8:2. Each dataset contains one continuous long time series, and we obtain input samples from the continuous long time series with a fixed length sliding window.

Table 6: Details of benchmark datasets. AR (anomaly ratio) represents the abnormal proportion of the whole dataset.

| Benchmark | Source | Variate Number | Window Length | Training & Validation (Unlabeled) | Test (Labeled) | AR (%) |
|---|---|---|---|---|---|---|
| MSL | NASA Space Sensors | 55 | 100 | 58,317 | 73,729 | 10.5 |
| SMAP | NASA Space Sensors | 25 | 100 | 135,183 | 427,617 | 12.8 |
| PSM | eBay Server Machine | 25 | 100 | 132,481 | 87,841 | 27.8 |
| SMD | Internet Server Machine | 38 | 100 | 708,405 | 708,420 | 4.2 |
| SWaT | Infrastructure System | 51 | 100 | 495,000 | 449,919 | 12.1 |
| NeurIPS-TS-SWAN | Space (Solar) Weather | 38 | 36 | 60,000 | 60,000 | 32.6 |
| NeurIPS-TS-GECCO | Water Quality for IoT | 9 | 90 | 69,260 | 69,261 | 1.1 |

## B  IMPLEMENTATION DETAILS

**Implementation details**  For real-world benchmarks, we follow the well-established protocol in Shen et al. (2020); Xu et al. (2021) and adopt a length-100 sliding window to obtain input samples for all datasets. For NeurIPS-TS benchmarks, we follow Lai et al. (2021); Yang et al. (2023) to set the sliding window length as 36 for NeurIPS-TS-SWAN and 90 for NeurIPS-TS-GECCO. We label the time points as anomalies if their anomaly scores (Equation 5) are larger than a certain threshold $\delta$. And the protocol to determine the threshold $\delta$ is introduced in Appendix E. We also adopt the widely-used adjustment strategy (Xu et al., 2018; Su et al., 2019; Shen et al., 2020; Xu et al., 2021; Yang et al., 2023): if a time point in a certain successive abnormal segment is detected, all anomalies in this abnormal segment are viewed to be correctly detected. This strategy is justified from the observation that an abnormal time point will cause an alert and further make the whole segment noticed in real-world applications. As the default model configurations, AnomalyTCN contains 1 layer ($L = 1$). And we set the channel number $D = 8$. In our asymmetric dual-branch settings, one branch is of a dense depth-wise convolution with kernel size 7, and the other branch is of a dilated depth-wise convolution with kernel size 3 and dilation ratio 3. Each branch contains a successive point-wise convolution module and these successive point-wise convolution modules in two branches do not share the weights. We use the ADAM (Kingma & Ba, 2014) optimizer with an initial learning rate of $10^{-4}$. The training process is early stopped within 5 epochs with the batch size of 128. All the experiments are implemented in Pytorch (Paszke et al., 2019) with a single NVIDIA A100 40GB GPU. We provide the analysis of hyper-parameter sensitivity in Appendix F.

**Pseudo-code**  We present the pseudo-code of AnomalyTCN in Algorithm 1.

## C  FULL RELATED WORK

### C.1  CONTRASTIVE-BASED TIME SERIES ANOMALY DETECTION

The classic reconstruction-based methods suffer from the performance bottleneck for they are easier to be interfered by anomalies during the training phase (Yang et al., 2023). And the reconstruction

error is also not good enough to be an anomaly criterion (Xu et al., 2021). To address these issues, some contrastive-based methods are proposed. Their insight is that, since normal points usually share the similar latent patterns, the representation discrepancy under different views for normal points are less than that for anomalies. Thus they adopt the representation discrepancy as a better anomaly detection criterion. For example, Anomaly Transformer (Xu et al., 2021) adopts the association discrepancy as a complementary to the reconstruction error, resulting in a hybrid anomaly criterion. DCdetecor (Yang et al., 2023) directly removes the reconstruction task. It proposes a dual-attention structure and adopts a contrastive representation task. Given an input, it calculates two attention maps from two different views and detects anomalies by the larger attention discrepancy. But these two methods bring severe computation costs due to the quadratic complexity in their attention computation. To improve efficiency, we design an attention-free contrastive-based method in this paper.

## C.2 MULTI-BRANCH CONVOLUTION STRUCTURE

The idea of multi-branch convolution can be traced back to the early exploration in CV, where the family of Inception networks (Szegedy et al., 2014; 2015; 2016; Chollet, 2016; Liu et al., 2020) proposes a multi-branch convolution structure with different kernel sizes to aggregate features under different receptive fields. In 2020s, with the proposal of structural re-parameterization (Ding et al., 2021), the idea of multi-branch convolution is revitalized. RepLKnet (Ding et al., 2022) proposes that a dual-branch convolution structure with a large kernel and a small kernel can effectively enlarge the receptive fields. Following it, SLaK (Liu et al., 2022a) further proposes a tri-branch convolution structure consisting of two rectangular large kernel and a square small kernel. And UnirepLKnet (Ding et al., 2023) proposes a multi-branch convolution structure with five different dilation ratios and kernel sizes. These CV methods tend to aggregate the representation outputs from different branches to obtain a more informative new representation. But in this paper, we take a novel and opposite perspective on the multi-branch convolution structure. We distinguish the representation discrepancy between each branch and further use it as a distinguishable criterion for time series anomaly detection.

## C.3 UNSUPERVISED TIME SERIES ANOMALY DETECTION

There are various unsupervised time series anomaly detection methods based on different learning tasks. As the early studies, the density-estimation methods, such as LOF (Breunig et al., 2000), COF (Tang et al., 2002), DAGMM (Zong et al., 2018) and MPPCACD (Yairi et al., 2017), detect the anomalies by calculating the local density for outlier determination. And the clustering-based methods like SVDD, Deep SVDD, THOC and ITAD (Tax & Duin, 2004; Ruff et al., 2018; Shen et al., 2020; Shin et al., 2020) detect the anomalies by calculating the distance to cluster center.

With the rapid development of deep learning backbones, the reconstruction and autoregression tasks have become popular because they are easier to adapt to different kinds of deep learning backbones like LSTM (Hochreiter & Schmidhuber, 1997), VAE (Kingma & Welling, 2013) and GAN (Goodfellow et al., 2014). The reconstruction-based methods are ideally trained to only reconstruct the normal samples, and thereby detect the anomalies by larger reconstruction errors. And some of the representative reconstruction-based methods are LSTM-VAE (Park et al., 2018), OmniAnomaly (Su et al., 2019), InterFusion (Li et al., 2021), D3R (Wang et al., 2023), f-AnoGAN (Schlegl et al., 2019), MAD-GAN (Li et al., 2019) and BeatGAN (Zhou et al., 2019). Similarly, the autoregression-based models are ideally trained to only predict the future points of normal samples and detect the anomalies by larger prediction errors (Anderson & Kendall, 1976; Hundman et al., 2018; Tariq et al., 2019). And in recent years, the combination of the popular reconstruction task with the powerful general time series backbones (Luo & Wang, 2024; Tian Zhou, 2023; Wu et al., 2023; Bian et al., 2024) has further contributed to the rapid progress of the time series anomaly detection.

However, these classic reconstruction-based methods suffer from the performance bottleneck for they are easier to be interfered by anomalies during the training phase (Yang et al., 2023). And the reconstruction error is also not good enough to be an anomaly criterion (Xu et al., 2021). To address these issues, some contrastive-based methods are proposed. Their insight is that, since normal points usually share the similar latent patterns, the representation discrepancy under different views for normal points are less than that for anomalies. Thus they adopt the representation discrepancy as a better anomaly detection criterion. For example, Anomaly Transformer (Xu et al., 2021) adopts the association discrepancy as a complementary to the reconstruction error, resulting in a hybrid anomaly criterion. DCdetecor (Yang et al., 2023) directly removes the reconstruction task. It proposes a

dual-attention structure and adopts a contrastive representation task. Given an input, it calculates two attention maps from two different views and detects anomalies by the larger attention discrepancy. Although achieving better performance, these two methods bring severe computation costs due to the quadratic complexity in their attention computation. To provide a more efficient contrastive-based method, we design an attention-free solution in this paper.

### C.4 CONTRASTIVE LEARNING IN COMPUTER VISION AND TIME SERIES ANOMALY DETECTION

In computer vision (CV), contrastive learning is proposed for pre-training, where a model is trained to learn an invariant representation for each input image under different data argumentations. Then these learned representations will be transfer to CV downstream tasks like classification, object detection and segmentation. The goal of contrastive learning is to minimize the representation discrepancy. But this goal is not necessarily consistent with downstream tasks'. For example, embedding all inputs as a constant can achieve the goal of contrastive learning but such representations can not be used for downstream tasks, resulting in trivial and collapsing solutions (i.e., totally fail in many CV tasks).

To avoid collapsing solutions, the early studies foucs on defining the <positive, negative> sample pairs, of which the common method is to define the data arguementations of same input as positive samples and others are negative samples. And the goal of contrastive learning is also refined to learn a representation space where positive samples are close to each other while negative ones are far apart (Wu et al., 2018; Ye et al., 2019). Furthermore, MoCo (He et al., 2019; Chen et al., 2020c; 2021) and SimCLR (Chen et al., 2020a;b) improve the performance with momentum backbones or additional predictor layers. In addition to above common definition, <positive, negative> sample pairs can be defined in other ways. For example, van den Oord et al. (2018) defines them with predictive coding and adopts an infoNCE loss to further avoid collapsing solutions. And Tian et al. (2019) defines positive samples as different views of the same input.

On the other hand, some following studies further prove that negative samples are not necessary to avoid model collapsing. BYOL (Grill et al., 2020) and SimSiam (Chen & He, 2020) simplifies the framework of contrastive learning and reveals that the asymmetric structure design and the stop-gradient operation help to avoid collapsing solutions without the need of negative samples. And in time series anomaly detection domain, our AnomalyTCN and the previous state-of-the-art DCdetector (Yang et al., 2023) both belong to this category that does not require negative samples.

Contrastive learning has been introduced into time series anomaly detection in recent years (Yang et al., 2023; Xu et al., 2021), which detects anomalies by the representation discrepancy. And we also find that, compared with CV domain, contrastive-based methods are less likely to collapse in time series anomaly detection. This may attribute to the different properties of CV tasks and time series anomaly detection tasks. The goal of contrastive learning is to learn the representation discrepancy (or conversely to learn the invariant representations) and may be inconsistent with the needs of some CV downstream tasks, therefore leading to the collapsing solutions in many CV cases. But in time series anomaly detection, the learned representation discrepancy is naturally a good anomaly criterion, making contrastive learning well suited for the needs of time series anomaly detection tasks. Therefore, contrastive learning is less likely to collapse in time series anomaly detection, which can ensure the performance stability and encourage further exploration in this direction.

## D  SHOWCASES ANALYSIS

To provide an intuitive comparision, we provide some visualization in Figure 4 to show how our AnomalyTCN works on different types of anomalies. We follow Lai et al. (2021) to generate the synthetic univariate time series data with five different types of anomalies. The anomalies include two point-wise anomalies (global point and contextual point anomalies) and three pattern-wise anomalies (shapelet, seasonal and trend anomalies). As shown in Figure 4, our AnomalyTCN can robustly detect various types of anomalies, demonstrating its great potential for various kinds of real-world applications. Besides, our pure contrastive-based anomaly criterion is more distinguishable because the anomaly scores for abnormal points are significantly higher in all cases, which can clearly distinguish the rare abnormal points from plenty of normal points and thus helps to better detect the anomalies. As a comparison, the hybrid contrastive-based and reconstruction-based criterion in Anomaly Transformer (2021) is less distinguishable and less robust. Due to the disruption from the less-robust reconstruction part, it will even wrongly produce relatively higher anomaly scores in

non-anomalous areas, leading to its failure in some detection cases. Above comparison highlights the performance superiority of our pure contrastive-based solution than the hybrid ones, therefore proving the soundness of our solution to remove the classic but less-robust reconstruction tasks and only adopt the contrastive representation tasks.

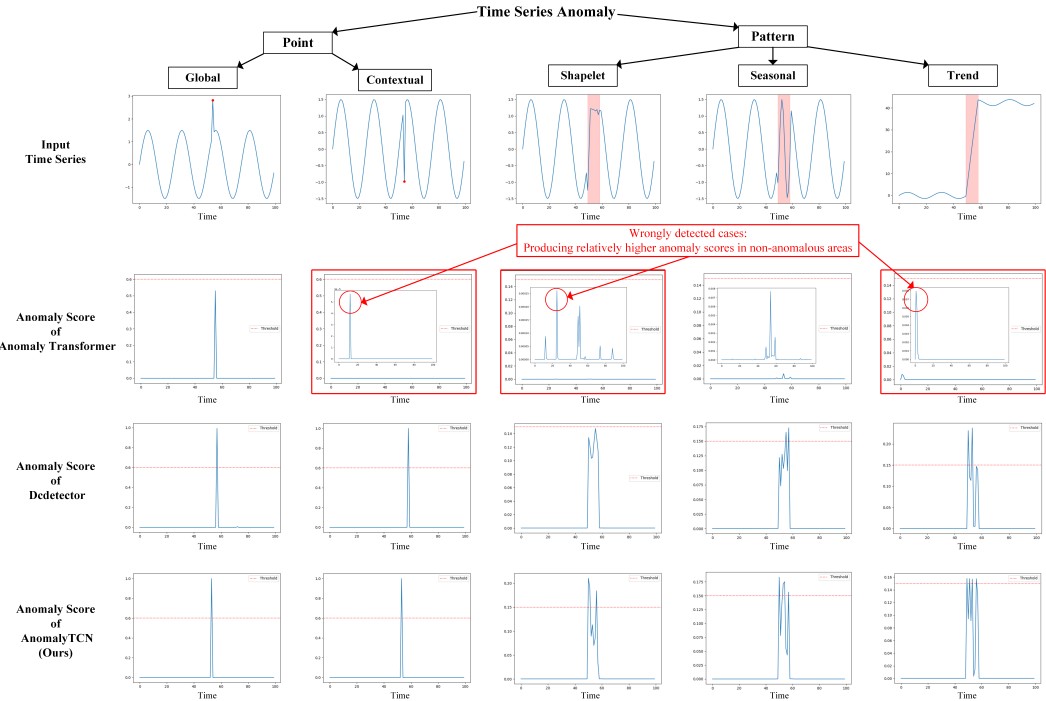

Figure 4: Visualization under different types of anomalies. We compare the anomaly scores among AnomalyTCN, DCdetector and Anomaly Transformer. The point-wise anomalies are marked by red points and the pattern-wise ones are in red segments. The wrongly detected cases (which produce relatively higher anomaly scores in non-anomalous areas) are bounded by red boxes.

## E    PROTOCOL OF THRESHOLD SELECTION

We follow the proctocol of Anomaly Transformer (2021) to decide the threshold $\delta$. The threshold $\delta$ is determined to make $r$ proportion data in the validation subset labeled as anomalies. Here is the selection procedure:

- After the training phase, we apply the model to the validation subset (without label) and obtain the anomaly scores (Equation 5) of all time points.
- We count the frequency of the anomaly scores in the validation subset. It is observed that the distribution of anomaly scores is separated into two clusters. We find that the cluster with a larger anomaly score contains $r$ time points. And for our model, $r$ is closed to 0.5% or 1% (Table 7).
- Due to the size of the test subset being still inaccessible in real-world applications, we have to fix the threshold as a fixed value $\delta$, which can gaurantee that the anomaly scores of $r$ time points in the validation set are larger than $\delta$ and thus detected as anomalies.

In real-world applications, the number of selected anomalies is always decided up to human resources. Under this consideration, setting the number of detected anomalies by the ratio $r$ is more practical and easier to decide according to the available resources.

And directly setting the $\delta$ is also an option. According to the intervals in Table 7, we can fix the $\delta$ as 0.01 for the SWaT and SMD datasets, 0.1 for MSL, and 0.5 for PSM and SMAP datasets. And we compare model performance under these two protocols. As shown in Table 8, directly setting the $\delta$ can achieve a quite close performance to setting $r$.

---

**Algorithm 1** Overall Structure of AnomalyTCN.

---

**Input:** $\mathbf{X} \in \mathbb{R}^{B \times T \times M}$: input time series; $B$: batch size; $T$, $M$ and $D$: the sizes of temporal, variate and feature dimemsions.

**Layer params:** `Embedding`: Embedding layer for input (implemented by point-wise convolution); `Conv1`: Dense convolution branch; `Conv2`: Dilated convolution branch.

1: ▷ Input embedding module.
2: $\mathbf{X} = \texttt{InstanceNorm}(\mathbf{X})$       ▷ $\mathbf{X} \in \mathbb{R}^{B \times T \times M}$
3: $\mathbf{X} = \mathbf{X}.\texttt{Permute}$       ▷ $\mathbf{X} \in \mathbb{R}^{B \times M \times T}$
4: $\mathbf{X} = \texttt{Reshape}(\mathbf{X})$       ▷ $\mathbf{X} \in \mathbb{R}^{(BM) \times 1 \times T}$
5: $\mathbf{X}_{emb} = \texttt{Embedding}(\mathbf{X})$       ▷ $\mathbf{X}_{emb} \in \mathbb{R}^{(BM) \times D \times T}$
6: ▷ Dual-branch convolution structure.
7: $\mathbf{S} = \texttt{Softmax}\Big(\texttt{Conv1}(\mathbf{X}_{emb}), \texttt{dims=-1}\Big)$       ▷ $\mathbf{S} \in \mathbb{R}^{(BM) \times D \times T}$
8: $\mathbf{P} = \texttt{Softmax}\Big(\texttt{Conv2}(\mathbf{X}_{emb}), \texttt{dims=-1}\Big)$       ▷ $\mathbf{P} \in \mathbb{R}^{(BM) \times D \times T}$
9: $\mathbf{P} = \mathbf{P}/\texttt{Broadcast}\Big(\texttt{Sum}(\mathbf{P}, \texttt{dim=-2})\Big)$       ▷ Rescaled $\mathbf{P} \in \mathbb{R}^{(BM) \times D \times T}$
10: $\mathbf{S} = \texttt{Reshape}(\mathbf{S})$       ▷ $\mathbf{S} \in \mathbb{R}^{B \times M \times D \times T}$
11: $\mathbf{P} = \texttt{Reshape}(\mathbf{P})$       ▷ $\mathbf{P} \in \mathbb{R}^{B \times M \times D \times T}$
12: Return $\mathbf{P}$ and $\mathbf{S}$ for contrastive discrepancy learning.

---

Table 7: Statistical results of anomaly score distribution on the validation subset. We count the number of time points with corresponding values in several intervals.

| Anomaly Score Interval | SMD | MSL | SMAP | SWaT | PSM |
|---|---|---|---|---|---|
| Number of total time points | 141681 | 11664 | 27037 | 99000 | 26497 |
| Number of time points in cluster1 | 140548 | 11547 | 26664 | 98535 | 26047 |
| Number of time points in cluster2 | 1133 | 117 | 373 | 465 | 450 |
| Ratio of cluster2 | 0.80% | 1.01% | 1.38% | 0.47% | 1.7% |

Table 8: Model performance under two protocols of threshold selection. A higher P, R, F1 (%) means better performance.

| Dataset | SMD | | | MSL | | | SMAP | | | SWaT | | | PSM | | |
|---|---|---|---|---|---|---|---|---|---|---|---|---|---|---|---|
| Metric | P | R | F1 | P | R | F1 | P | R | F1 | P | R | F1 | P | R | F1 |
| Setting $\delta$ | 86.86 | 92.93 | 89.79 | 92.43 | 99.71 | 95.93 | 93.67 | 99.58 | 96.53 | 93.28 | 100.00 | 96.53 | 97.07 | 97.62 | 97.35 |
| Setting $r$ | 86.88 | 92.93 | 89.80 | 92.76 | 99.71 | 96.11 | 93.67 | 99.58 | 96.53 | 93.33 | 100.00 | 96.55 | 97.08 | 98.83 | 97.95 |

Table 9: Model performance under different channel number $D$. A higher P, R, F1 (%) means better performance.

| Dataset | MSL | | | SMAP | | | SWaT | | | PSM | | |
|---|---|---|---|---|---|---|---|---|---|---|---|---|
| Metric | P | R | F1 | P | R | F1 | P | R | F1 | P | R | F1 |
| $D = 8$ | 92.76 | 99.71 | 96.11 | 93.67 | 99.58 | 96.53 | 93.33 | 100.00 | 96.55 | 97.08 | 98.83 | 97.95 |
| $D = 16$ | 92.76 | 99.71 | 96.11 | 93.67 | 99.48 | 96.49 | 93.62 | 99.06 | 96.26 | 97.08 | 98.36 | 97.72 |
| $D = 32$ | 92.64 | 98.92 | 95.68 | 93.65 | 99.30 | 96.39 | 93.22 | 99.59 | 96.30 | 97.00 | 98.86 | 97.92 |
| $D = 64$ | 92.62 | 98.92 | 95.66 | 92.98 | 98.36 | 95.59 | 92.96 | 98.88 | 95.83 | 96.96 | 97.36 | 97.16 |

Table 10: Model performance under different number of layers $L$. A higher P, R, F1 (%) means better performance.

| Dataset | MSL | | | SMAP | | | SWaT | | | PSM | | |
|---|---|---|---|---|---|---|---|---|---|---|---|---|
| Metric | P | R | F1 | P | R | F1 | P | R | F1 | P | R | F1 |
| $L = 1$ | 92.76 | 99.71 | 96.11 | 93.67 | 99.58 | 96.53 | 93.33 | 100.00 | 96.55 | 97.08 | 98.83 | 97.95 |
| $L = 2$ | 92.60 | 97.24 | 94.86 | 93.65 | 99.36 | 96.42 | 93.22 | 100.00 | 96.49 | 97.07 | 98.70 | 97.88 |
| $L = 3$ | 92.72 | 99.42 | 95.95 | 93.66 | 99.46 | 96.47 | 93.47 | 100.00 | 96.62 | 97.03 | 98.08 | 97.55 |
| $L = 4$ | 92.70 | 98.92 | 95.71 | 93.65 | 99.29 | 96.39 | 93.35 | 100.00 | 96.56 | 97.07 | 98.49 | 97.77 |
| $L = 5$ | 92.68 | 98.92 | 95.70 | 93.66 | 99.36 | 96.42 | 93.33 | 100.00 | 96.55 | 97.05 | 98.20 | 97.62 |

# F  PARAMETER SENSITIVITY AND MORE ABLATION STUDIES

## F.1  STUDY ON MODEL PARAMETER

To see whether AnomalyTCN is sensitive to the choice of model parameters, we conduct experiments with varying model parameters, including channel number ranging from $D \in \{8, 16, 32, 64\}$ and the number of layers ranging from $L \in \{1, 2, 3, 4, 5\}$. As shown in Table 9 and 10, our model is robust to the choice of model parameters. Considering both performance and efficiency, we set $L = 1$ and $D = 8$ in main experiments by default.

## F.2  IMPACT OF INSTANCE NORMALIZATION

Instance normalization (2021) has already become a widely-used technique in various latest time series models, which can make the training process more stable. And we conduct experiments to study the impact of instance normalization. As shown in Table 11, instance normalization slightly improves the performance of our model. Therefore, we adopt the instance normalization in main experiments by default.

Table 11: Ablation about instance normalization. *+IN* means with instance normalization. *-IN* means without instance normalization. A higher P, R, F1 (%) means better performance.

| Dataset | MSL | | | SMAP | | | SWaT | | | PSM | | |
|---|---|---|---|---|---|---|---|---|---|---|---|---|
| Metric | P | R | F1 | P | R | F1 | P | R | F1 | P | R | F1 |
| +IN | 92.76 | 99.71 | 96.11 | 93.67 | 99.58 | 96.53 | 93.33 | 100.00 | 96.55 | 97.08 | 98.83 | 97.95 |
| -IN | 92.62 | 98.92 | 95.66 | 93.66 | 99.46 | 96.47 | 93.26 | 100.00 | 96.51 | 97.01 | 97.92 | 97.47 |

Table 12: Model performance under different window lengths. A higher P, R, F1 (%) means better performance.

| Dataset | MSL | | | SMAP | | | SWaT | | | PSM | | |
|---------|-----|-----|-----|------|-----|-----|------|-----|-----|-----|-----|-----|
| Metric | P | R | F1 | P | R | F1 | P | R | F1 | P | R | F1 |
| 50 | 91.32 | 86.23 | 88.70 | 92.57 | 98.70 | 95.54 | 92.93 | 100.00 | 96.34 | 97.14 | 98.60 | 97.86 |
| 100 | 92.76 | 99.71 | 96.11 | 93.67 | 99.58 | 96.53 | 93.33 | 100.00 | 96.55 | 97.08 | 98.83 | 97.95 |
| 150 | 92.61 | 99.33 | 95.85 | 93.98 | 98.31 | 96.10 | 93.18 | 99.16 | 96.08 | 96.64 | 98.74 | 97.68 |
| 200 | 92.07 | 97.89 | 94.89 | 93.91 | 98.44 | 96.12 | 93.06 | 99.90 | 96.36 | 96.86 | 98.55 | 97.70 |
| 250 | 92.50 | 97.72 | 95.04 | 93.49 | 97.17 | 95.29 | 93.18 | 99.97 | 96.46 | 97.40 | 98.66 | 98.02 |
| 300 | 94.40 | 92.14 | 93.26 | 93.07 | 96.96 | 94.97 | 92.43 | 94.29 | 93.35 | 97.22 | 98.68 | 97.94 |

Table 13: Model performance under different statistical distances in loss function. A higher P, R, F1 (%) means better performance.

| Dataset | MSL | | | SMAP | | | SWaT | | | PSM | | |
|---------|-----|-----|-----|------|-----|-----|------|-----|-----|-----|-----|-----|
| Metric | P | R | F1 | P | R | F1 | P | R | F1 | P | R | F1 |
| JSD | 92.85 | 97.52 | 95.13 | 93.65 | 99.38 | 96.43 | 93.09 | 98.57 | 95.75 | 98.10 | 97.47 | 97.79 |
| CE | 92.11 | 94.00 | 93.05 | 93.14 | 69.45 | 79.57 | 93.29 | 100.00 | 96.53 | 97.14 | 99.14 | **98.13** |
| Wasserstein | 91.33 | 94.30 | 92.79 | 91.54 | 56.48 | 69.86 | 93.34 | 95.46 | 94.39 | 97.03 | 98.52 | 97.77 |
| L2 | 76.89 | 53.95 | 63.41 | 93.84 | 96.80 | 95.30 | 91.83 | 90.58 | 91.20 | 97.26 | 96.17 | 96.71 |
| Ours | 92.76 | 99.71 | **96.11** | 93.67 | 99.58 | **96.53** | 93.33 | 100.00 | **96.55** | 97.08 | 98.83 | 97.95 |

### F.3 IMPACT OF WINDOW LENGTH

In time series anomaly detection, each dataset contains one continuous long time series, and we obtain input samples from the continuous long time series with a fixed length sliding window. Since a single time point is not enough to be considered as a sample, using a sliding window to split time series into instances is very important. And therefore the window length is a significant hyper-parameter. As shown in Table 12, our AnomalyTCN is robust to different window lengths. In main experiments, we follow the well-established protocol in Shen et al. (2020); Xu et al. (2021) to set the window length as 100 for real-world datasets, which can provide a fair comparison for all baselines. Meanwhile, since a larger window length will lead to a heavier computational cost, setting the window length as 100 is also a good choice that considers both performance and efficiency.

### F.4 STUDY ON DIFFERENT STATISTICAL DISTANCES IN LOSS FUNCTION

We use different statistical distances to calculate the discrepancy between the two representations of our dual-branch structure, and the results are shown in Table 13. The statistical distances we used in this study are : Symmetrized Kullback–Leibler Divergence (Ours), Jensen–Shannon Divergence (JSD), Wasserstein Distance (Wasserstein), Cross-Entropy (CE) and L2 Distance (L2).

Our proposed loss function with Symmetrized Kullback–Leibler Divergence still achieves the best performance. We find that the JSD, CE and Wasserstein can provide fairly good results on some datasets. But they fail to work well on all datasets. Therefore, we use the proposed loss function with Symmetrized Kullback–Leibler Divergence in main experiments.

### F.5 ROC CURVE

The predefined threshold proportion $r$ is a hyperparameter which has an impact on determining whether a time point is an anomaly or not. To study its impact, we provide the receiver operator

characteristic (ROC) curve in Figure 5. And we also adopt a representative contrastive-based model (DCdetector (2023)) and a representative reconstructed-based model (TimesNet (2023)) for comparison. As shown in Figure 5, our AnomalyTCN has the highest AUC values in all cases and performs well in the false-positive and true-positive rates under various pre-selected thresholds. The result verifies our robustness to the choice of pre-selected thresholds, which is important for real-world applications.

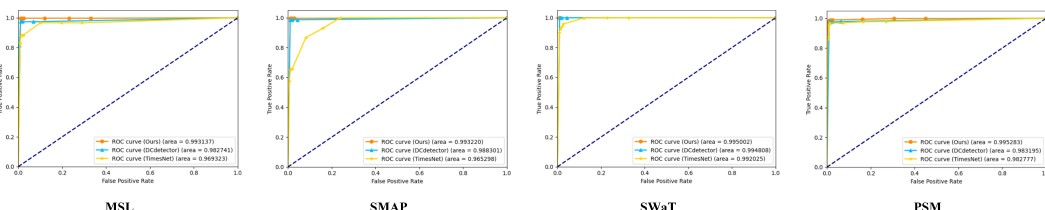

Figure 5: ROC curves (horizontal-axis: false-positive rate; vertical-axis: true-positive rate). A higher AUC value (area under the ROC curve) indicates a better performance. The predefined threshold proportion $r$ is in $\{0.5\%, 1.0\%, 1.5\%, 2.0\%, 10\%, 20\%, 30\%\}$.

## G  MORE STUDIES ON MODEL EFFICIENCY

**Impact of Embedding Methods**  In main experiments, our AnomalyTCN adopts the idea of variate-independence to embed the time series variate-independently, which is implemented by $\text{Embedding} : \mathbb{R}^{M \times 1 \times T} \mapsto \mathbb{R}^{M \times D \times T}$. This embedding maintains the variate dimension in the embedded series and leads to more memory usage. By contrast, some traditional anomaly detection methods adopt the variate-mixing embedding, which simply embed the $M$ variates in to a $D$-dimensional vector at each time step with $\text{Embedding} : \mathbb{R}^{T \times M} \mapsto \mathbb{R}^{T \times D}$. This more light-weight embedding discards the variate dimension in the embedded series and helps to save memory usage.

To study the impact of different embedding mthods and to test whether our efficiency can be further improved by the variate-mixing embedding, we also compare our AnomalyTCN under these two different embedding mthods. In variate-independent embedding, the default $D$ is set as $8$. In variate-mixing embedding, since we embed the information of the whole $M$ variates into a single $D$-dimensional vector, the default $D$ should be larger ($D = 128$) to avoid information loss. And the results are shown in Table 14. When using variate-mixing embedding, our AnomalyTCN can still achieve comparable performance and bring better efficiency. Some discussions are as follows.

The idea of variate-independent embedding is recently proposed in time series forecasting area (Nie et al., 2023) and it is proven that the variate-independent embedding performs much better than the variate-mixing embedding in forecasting tasks (Luo & Wang, 2024; Nie et al., 2023). But in time series anomaly detection tasks, we observe that the variate-mixing embedding doens't lead to severe performance degradation and can achieve comparable performance to the variate-independent embedding. This may be attributed to the difference between these two tasks. In forecasting tasks, we intend to predict the future of each variate respectively. Therefore maintaining the independence of the variates is important. But in anomaly detection tasks, the goal is to detect whether there are anomalies at each time step. This process is carried out jointly among all variates. As a result, aggregating and mixing the variate information in advance is also an acceptable operation. Therefore, variate-mixing embedding doesn't lead to severe performance degradation in time series anomaly detection tasks, and the efficiency of AnomalyTCN can be further improved with the help of variate-mixing embedding. But given that variate-independence has become the mainstreaming choice in time series community, we still adopt variate-independent embedding in main experiments to provide a fair comparison.

Table 14: Model efficiency of AnomalyTCN under two embedding methods. *F1*, *Time* and *Mem* means the F1-score (%), Averaged running time of 100 iterations (s/iter) and Peak memory usage (GB). A higher *F1* means better performance and a lower *Time* and *Mem* indicate better efficiency.

| Dataset | SMD | | | MSL | | | SMAP | | | SWaT | | | PSM | | | NIPS-TS-SWAN | | | NIPS-TS-GECCO | | |
|---|---|---|---|---|---|---|---|---|---|---|---|---|---|---|---|---|---|---|---|---|---|
| Embedding method | F1 | Time | Mem | F1 | Time | Mem | F1 | Time | Mem | F1 | Time | Mem | F1 | Time | Mem | F1 | Time | Mem | F1 | Time | Mem |
| Variate-independent | 89.80 | 0.015 | 17.5 | 96.11 | 0.020 | 24.4 | 96.53 | 0.010 | 14.3 | 96.55 | 0.021 | 22.8 | 97.95 | 0.010 | 14.3 | 74.1 | 0.008 | 8.9 | 55.0 | 0.007 | 8.2 |
| Variate-mixing | 89.43 | 0.010 | 8.5 | 96.16 | 0.010 | 8.5 | 96.55 | 0.010 | 8.5 | 96.52 | 0.010 | 8.5 | 97.79 | 0.010 | 8.5 | 73.3 | 0.008 | 3.1 | 51.9 | 0.009 | 8.5 |

## H    FULL RESULTS FOR NEURIPS-TS BENCHMARKS

Due to the space limitation of the main text, we place the full results of NeurIPS-TS benchmarks in Table 15, which includes more metrics and more baselines.

Table 15: Full results on NeurIPS-TS benchmarks. A higher P, R, F1 (%) means better performance, and the best ones are in **bold**. The models are ranked from lowest to highest based on the average performance.

| Dataset | NIPS-TS-GECCO | | | NIPS-TS-SWAN | | | Avg F1 |
|---|---|---|---|---|---|---|---|
| Metric | P | R | F1 | P | R | F1 | (%) |
| OCSVM* | 2.1 | 34.1 | 4.0 | 19.3 | 0.1 | 0.1 | 2.05 |
| MatrixProfile | 4.6 | 18.5 | 7.4 | 16.7 | 17.5 | 17.1 | 12.25 |
| GBRT | 17.5 | 14.0 | 15.6 | 44.7 | 37.5 | 40.8 | 28.20 |
| LSTM-RNN | 34.3 | 27.5 | 30.5 | 52.7 | 22.1 | 31.2 | 30.85 |
| Autoregression | 39.2 | 31.4 | 34.9 | 42.1 | 35.4 | 38.5 | 36.70 |
| OCSVM | 18.5 | 74.3 | 29.6 | 47.4 | 49.8 | 48.5 | 39.05 |
| IForest* | 39.2 | 31.5 | 39.0 | 40.6 | 42.5 | 41.6 | 40.30 |
| THoC | 22.7 | 32.1 | 27.4 | 76.6 | 48.3 | 59.2 | 43.30 |
| AutoEncoder | 42.4 | 34.0 | 37.7 | 49.7 | 52.2 | 50.9 | 44.30 |
| Anomaly Transformer | 25.7 | 28.5 | 27.0 | 90.7 | 47.4 | 62.3 | 44.65 |
| IForest | 43.9 | 35.3 | 39.1 | 56.9 | 59.8 | 58.3 | 48.70 |
| GPT4TS | 32.6 | 49.5 | 39.3 | 95.9 | 53.7 | 68.8 | 54.05 |
| D3R | 51.3 | 37.7 | 43.5 | 86.1 | 55.0 | 67.1 | 55.30 |
| TimesNet | 52.1 | 38.4 | 44.2 | 96.5 | 56.2 | 71.0 | 57.60 |
| aLLM4TS | 46.9 | 49.2 | 48.0 | 96.2 | 53.4 | 68.7 | 58.35 |
| ModernTCN | 59.7 | 38.4 | 46.7 | 96.7 | 55.3 | 70.4 | 58.55 |
| DCdetector | 38.3 | 59.7 | 46.6 | 95.5 | 59.6 | 73.4 | 60.00 |
| AnomalyTCN (Ours) | 45.7 | 69.2 | **55.0** | 97.8 | 59.6 | **74.1** | **64.55** |

## I    UNIVARIATE TIME SERIES ANOMALY DETECTION IN UCR DATASET

We conduct the univariate time series anomaly detection tasks on UCR Dataset (Keogh et al., 2021). The whole UCR dataset contains 250 sub-datasets, ranging in length from 6,684 to 900,000 and covering various real-world scenarios. And each sub-dataset has only one anomaly segment. All these sub-datasets are pre-divided into training and test sets. We trained and tested separately for each of the sub-datasets and provide the average results in Table 16. The results show that our AnomalyTCN still achieves the state-of-the-art in this challenging benchmark.

Table 16: Results of anomaly detection in UCR. A higher P, R, F1 (%) means better performance, and the best ones are in **bold**. The models are ranked from lowest to highest based on the performance.

| Dataset | UCR | | |
|---|---|---|---|
| Metric | P | R | F1 |
| Anomaly Transformer | 60.41 | 100.00 | 75.32 |
| DCdetector | 61.62 | 100.00 | 76.25 |
| AnomalyTCN (Ours) | 62.88 | 100.00 | **77.21** |

