# OpenReview forum: "AnomalyTCN: Dual-branch Convolution with Contrastive Representation for Efficient Time Series Anomaly Detection"
_ICLR.cc/2025/Conference — ICLR 2025 Conference Withdrawn Submission_

### Official Review · Reviewer_itba · 2024-10-24

**Soundness:** 3
**Presentation:** 3
**Contribution:** 2
**Rating:** 5
**Confidence:** 4

**Summary:**

Time series anomaly detection is widely used in extensive real-world applications. Recently, various reconstruction-based methods are developed to handle anomaly detection problems. However, such methods are less-robust for they are easily interfered by the uncleanliness of training data. In order to solve the above problem, the author proposes AnomalyTCN as an  efficient and effective solution. Technically, a dual-branch convolution structure is proposed to produce different representations of the same input from two different views. Then the representation discrepancy of two convolution branches is calculated for contrastive learning. Experiments demonstrate that AnomalyTCN performs well in terms of both effectiveness and efficiency in anomaly detection.

**Strengths:**

1.	The author proposed an effective and efficient method for handling anomalous instances, achieving state-of-the-art results in anomaly detection.

2.	Clear charts and comprehensive experiments illustrate the structure presented in the paper and verify the effectiveness of the method.

**Weaknesses:**

1.	The contribution of this paper is relatively limited. The main idea proposed here, which aims to improve performance through contrastive learning, has already been introduced in DCdetector. The training framework and loss function of this paper are quite similar to those in DCdetector. The contribution of this work lies in the utilization of convolution and the improvement of the contrastive learning speed, which is relatively limited.

2.	More explanation is needed on why using a contrastive-based approach can better differentiate between normal points and anomalous points. Will the presence of anomalous data in the training set also influence the training process of contrastive learning, leading to smaller differences in multiple representations of the final anomalous instances?

3.	The robustness of AnomalyTCN can be further validated through more experiments. Different proportions of anomalies can be added to the training set to observe their influence on the anomaly detection performance of AnomalyTCN.

4.	The utilization of the proposed structure and the advantages of using convolution need further explanation. More explanation or experiments are required to illustrate why the convolution-based method can outperform the attention-based method in terms of effectiveness.

5.	Some experimental results contain labeling errors: For example, in the PSM dataset shown in Table 1, the F1-score of TimesNet is 97.47, which surpasses the F1-score of D3R. Therefore, the experimental results of TimesNet should be marked as underlined.

**Questions:**

1.	According to table 16, the author only compared the univariate anomaly detection performance of AnomalyTCN with Anomaly Transformer and DCdetector. Why are other baselines not included for performance comparison in this case?

2.	The paper mentions that when outliers are present in the training data, training based on reconstruction methods may suffer due to the presence of outliers, while contrastive learning-based methods perform relatively better. Could you provide some case studies with real datasets to support this idea? For example, certain time series can be visualized to show that anomaly patterns only influence reconstruction-based methods but do not affect contrastive learning-based approaches.

---

### Official Review · Reviewer_ULqR · 2024-10-28

**Soundness:** 2
**Presentation:** 3
**Contribution:** 2
**Rating:** 3
**Confidence:** 4

**Summary:**

To solve the efficiency issue in previous costly attention-based solutions, this paper proposes AnomalyTCN as a more efficient and effective solution. Technically, a dual-branch convolution structure is proposed to produce different representations of the same input from two different views. Then it calculates the representation discrepancy of these two convolution branches for contrastive learning and further derive a more distinguishable anomaly criterion based on this discrepancy, resulting in better detection performance.

**Strengths:**

S1. Time series anomaly detection is important to various domains.

S2. There are quite a few nice illustrations.

S3. This work focuses on an important problem that could have real-world applications.

S4. The figures and tables used in this work are clear and easy to read.

S5. Compared with other contrastive learning methods, this work significantly improves the efficiency of TSAD.

**Weaknesses:**

W1: Code or more detailed implementation details should be provided.

W2: This method has strong limitations. Can it solve other types of exceptions, such as Global point, Contextual point, Seasonal, Trend, Shapelet exceptions proposed in [1].

W3: The specific innovation points of the entire article are insufficient, mostly utilizing existing technologies.

W4: The evaluation metrics reported in the work are relatively limited. Can the performance of the algorithm be reflected on other diversified metrics. The point adjust operation has been proven by many articles to be incorrect and cannot reflect the actual performance of the algorithm [2].

W5: Why were the experiments in Table 3-5 not experimentally validated on all datasets？

[1] Revisiting time series outlier detection: Definitions and benchmarks.

[2] Towards a rigorous evaluation of time-series anomaly detection

**Questions:**

W1-W5

---

### Official Review · Reviewer_5TDW · 2024-10-30

**Soundness:** 2
**Presentation:** 3
**Contribution:** 1
**Rating:** 1
**Confidence:** 4

**Summary:**

This paper propose a simple framework AnomalyTCN for anomaly detection with pure convolution structure and dual-branch contrastive learning. The authors think that AnomalyTCN can provide a better balance between the performance and efficiency of anomaly detection.

**Strengths:**

1. This paper presents an efficient framework AnomalyTCN for anomaly detection.
2. This paper presents pure convolution structure for dual-branch contrastive learning.
3. The experimental findings illustrate the efficiency of this approach.

**Weaknesses:**

1. The claimed novelty in this paper says that "novelly propose a dual-branch pure convolution structure for contrastive-based time series anomaly detection". However, dual-branch contrastive-based time series anomaly detection is already proposed by DCdetector [1], and there are also pure convolution structure time series anomaly detection models, such as [2]. The contributions and novelty are limited. The authors should compare these prior works by experiments and provide a detailed comparison highlighting the key technical differences and innovations of their method.

   [1] DCdetector: Dual Attention Contrastive Representation Learning for Time Series Anomaly Detection. KDD 2023.

   [2] Explainable Abnormal Time Series Subsequence Detection Using Random Convolutional Kernels. DeLTA 2023.

2. As the efficiency by pure convolution structure is the main contribution, why the authors compare efficiency with only one Transformer based model? The authors should compare efficiency and effectiveness with other convolution structure time series anomaly detection models, such as TimesNet, ModernTCN and [2], as well as non-learning anomaly detection methods, such as [3] and [4]. The authors should use more metrics to compare efficiency (e.g. training time, inference time, memory usage, model parameters) across more models.

   [3] Matrix Profile XXIX: C22MP, Fusing catch 22 and the Matrix Profile to Produce an Efficient and Interpretable Anomaly Detector. ICDM 2023.

   [4] Matrix Profile XXIV: Scaling Time Series Anomaly Detection to Trillions of Datapoints and Ultra-fast Arriving Data Streams. KDD 2022.

3. The main weakness of this paper is that the authors use the wrong evaluation metrics. The authors use the point adjustment (PA) for evaluation. Many works [2, 5, 6, 7] have demonstrated that PA can lead to faulty performance evaluations, where PA use true labels from the test datasets to adjust the outputs of models, and it is known that using PA can result in state-of-the-art performance even with random scores or random initialized non-trained models [6, 7], making it impossible to conduct a fair comparison and assess the effectiveness of the models. The showed effectiveness in this paper is flawed. The authors should use alternative evaluation metrics, such as original F1, ROC-AUC, PR-AUC, Aff Recall, Precision and F1 [6, 7], instead of PA. The authors should re-run their experiments with these reasonable metrics and discuss how it impacts their results and conclusions.

   [5] Current Time Series Anomaly Detection Benchmarks are Flawed and are Creating the Illusion of Progress. TKDE 2023.

   [6] Drift doesn't Matter: Dynamic Decomposition with Diffusion Reconstruction for Unstable Multivariate Time Series Anomaly Detection. NeurIPS 2023.

   [7] Local Evaluation of Time Series Anomaly Detection Algorithms. KDD 2022.

4. No codes available. I find the results in this paper are not reliable. From the original paper such as the baseline DCdetector, the F1 results are larger than the results in this paper. Can the authors explain what is the reason and provide the codes to evaluate the results and ensure reproducibility?

   -Provide a link to the code and data.

   -Explain in detail the experimental setup, including any preprocessing steps.

   -Clarify why the reported baseline results differ from the original papers.

   -Re-run the experiments with reasonable metrics.

**Questions:**

see weaknesses.

---

### Official Review · Reviewer_fL4T · 2024-11-03

**Soundness:** 3
**Presentation:** 3
**Contribution:** 2
**Rating:** 5
**Confidence:** 4

**Summary:**

This paper proposes a convolution-based model AnomalyTCN for time series anomaly detection, which can efficiently and effectively detect anomalies with the discrepancy between different channels. Specifically,  AnomalyTCN first feds the time series into a dense kernel-based convolution and another dilated one to derive different representations, and then detects anomalies through the discrepancy between these two representations because abnormal points can be skipped by the dilated kernel. Experiments on various time series anomaly detection datasets demonstrate the better balance of performance and efficiency of AnomalyTCN.

**Strengths:**

1. This paper proposes a novel convolution-based model for time series anomaly detection with contrastive discrepancy.

2. The proposed model achieves superior performance with low computation needs over several baselines.

**Weaknesses:**

1. There is also a probability that abnormal points are ignored after the dilated convolution and thus decrease the performance of AnomalyTCN. Is there any proof from the theoretical or experimental that the dense and dilated convolution is indeed inconsistent due to abnormal points?

2. Why not compare the latest contrastive-based time series anomaly detection method TFMAE[1]?

3. Many important contrastive-based time series anomaly detection methods are ignored in the related work such as TimeAutoAD[2] and CTAD[3].

[1] Temporal-Frequency Masked Autoencoders for Time Series Anomaly Detection

[2] Timeautoad: Autonomous anomaly detection with self-supervised contrastive loss for multivariate time series

[3] Contrastive time-series anomaly detection

**Questions:**

Please see in weaknesses.

---

### Official Review · Reviewer_z6Gw · 2024-11-03

**Soundness:** 2
**Presentation:** 2
**Contribution:** 2
**Rating:** 3
**Confidence:** 3

**Summary:**

This paper proposes a contrastive-based method for time series anomaly detection. Different from existing methods, this paper proposes a
 simple and lightweight pure convolution structure, which is more efficient and effective than the complicated attention mechanism. Although it is the first to use a convolution module in anomaly detection, the idea of this paper is not novel.

**Strengths:**

- The proposed method is more efficient than the existing methods.

- The experiment of this paper is extensive.

**Weaknesses:**

- The writing of this paper needs to be improved. For example, this paper spends two pages on the introduction, but the challenge that this paper aims to solve is still not very clear.

- The idea of this paper is trivial and lacks novelty. This paper proposes two branches of convolution, i.e., dense and dilated, to learn two views of time series. Then, the contrastive learning method is applied to learn a time series anomaly detection. However, this structure is widely used in other research areas, and it is trivial to apply this structure again in the application of time series anomaly detection.

- Some experiments show that the proposed method has very minor improvements on many datasets, e.g., SMD, SMAP, SWaT, PSM, and NIPS-TS-SWAN. With these minor improvements and the trivial of the proposed method, it makes the reader to wonder the effectiveness of the proposed method.

**Questions:**

See above weaknesses.

---

### Note · Authors · 2024-11-24

I have read and agree with the venue's withdrawal policy on behalf of myself and my co-authors.